# Characterisation of a Novel Complex Concentrated Alloy for Marine Applications

**DOI:** 10.3390/ma15093345

**Published:** 2022-05-06

**Authors:** Ioana-Cristina Badea, Ioana Csaki, Beatrice-Adriana Serban, Nicolae Constantin, Dumitru Mitrica, Marian Burada, Ioana Anasiei, Mihai Tudor Olaru, Andreea-Nicoleta Ghita, Ana-Maria Julieta Popescu

**Affiliations:** 1National R&D Institute for Non-Ferrous and Rare Metals, 102 Biruintei Blvd, 077145 Pantelimon, Romania; beatrice.carlan@imnr.ro (B.-A.S.); mburada@imnr.ro (M.B.); ianasiei@imnr.ro (I.A.); o.mihai@imnr.ro (M.T.O.); andreea.lupu9@yahoo.com (A.-N.G.); 2Engineering and Management of Metallic Materials Production Department, Faculty of Materials Science and Engineering, University Politehnica of Bucharest, 060042 Bucharest, Romania; ioana.apostolescu@upb.ro (I.C.); nctin2014@yahoo.com (N.C.); 3“Ilie Murgulescu” Institute of Physical Chemistry, Romanian Academy, 060021 Bucharest, Romania; popescuamj@yahoo.com

**Keywords:** complex concentrated alloys, corrosion resistance, transportation applications, materials design

## Abstract

Complex concentrated alloys (CCAs) are a new family of materials with near equimolar compositions that fluctuate depending on the characteristics and destination of the material. CCAs expand the compositional limits of the traditional alloys, displaying new pathways in material design. A novel light density Al_5_Cu_0.5_Si_0.2_Zn_1.5_Mg_0.2_ alloy was studied to determine the structural particularities and related properties. The alloy was prepared in an induction furnace and then annealed under a protective atmosphere. The resulted specimens were analysed by chemical, structural, mechanical, and corrosion resistance. The structural analyses revealed a predominant FCC and BCC solid solution structure. The alloy produced a compression strength of 500–600 MPa, comparable with conventional aluminium alloys. The corrosion resistance in 3.5% NaCl solution was 0.3424 mm/year for as-cast and 0.1972 mm/year for heat-treated alloy, superior to steel, making the alloy a good candidate for marine applications.

## 1. Introduction

The corrosion of metallic materials is a natural and inevitable phenomenon. This process causes losses in industrial activities estimated at one-third of the total losses in the world economy. Therefore, this phenomenon has been the base of numerous experimental studies aimed at preventing or slowing down the damage of metallic parts and equipment caused by the influence of the contact environment. As a result, the main preoccupations are the elaboration of materials with anticorrosive properties and the protection of the metals against corrosion [1].

Alloys with high resistance in corrosive environments contain a high concentration of elements that ensure corrosion resistance. However, this limits the selection of corrosion-resistant alloy compositions [2]. For example, major importance for the corrosion resistance of austenitic steels relates to the chromium content. Chromium facilitates the formation of a passive film, and molybdenum helps maintain it by promoting repassivation [3].

With the need to cover the demands of a society in continuous progress, efforts have resulted in the development of new materials with superior properties. The development of corrosion-resistant alloys with properties that exceed the limits of conventional alloys has made significant progress due to the so-called high entropy alloys (HEA), more generally called complex concentrated alloys (CCA).

The concept of CCA involves developing multicomponent alloys composed of at least 4 main elements at high concentrations. A specific feature of these types of alloys is the high configurational entropy that leads to the formation of preferential solid solution solutions [4].

Slow diffusion and severe lattice distortions are two other characteristics of complex alloys that directly influence their microstructures and properties. Thus, high hardness and strength, high corrosion resistance, oxidation, or wear are determined by the above-mentioned characteristics [5].

Alloys with a single solid solution phase are often the most desirable due to their superior corrosion resistance [6]. The large space of the compositions in which the multicomponent alloys are found offers the opportunity to achieve optimal passivation by choosing the appropriate composition [7,8]. The corrosion behavior of complex concentrated (or high entropy) alloys has been reported in numerous scientific papers. Various studies were based on the CoCrFeNi equimolar system. CoCrFeNi-Cu_x_ has also been studied, and the effects of increasing the Cu content on the microstructure and corrosion properties have been reported [9]. Besides, the addition of Al in the CoCrFeNi alloy has a major influence on the internal structure and mechanical characteristics of the alloy, considering the large atomic radius of aluminum [10]. This statement was reinforced by Tong et al. and Li et al. They studied the Al_x_CoCrCuFeNi alloy system and reported the significant effect of Al content on the phase transformations and mechanical properties of the alloy [5].

Research on the corrosion behavior of Al_0.1_CoCrFeNi alloy reveals that the values of corrosion resistance were higher compared to SS304 stainless steel. This can be explained by forming a strongly passivated surface layer due to the high Cr and Ni content [10].

The corrosion behavior of CCAs is generally influenced by such factors as the elements that enter the composition of the alloy, the environment in which the corrosion occurs, and the process parameters. So far, most of the CCA alloy studies have shown good corrosion resistance compared to stainless steel due to the high content of elements characterized by a strong passivating effect. The presence in the alloy of some metals that contain unpaired electrons confers unsaturation to the surface, so the catalytic activity and the adsorption of oxygen are favored [10,11,12].

The investigations regarding the development of a novel complex concentrated alloy based on the Al-Cu-Si-Zn-Mg system with emphasis on its corrosion resistance is the main topic of this study. The alloy was designed to have properties specific to the transport industry. The alloy should replace steel in various structures due to the high mechanical resistance, low density, and improved corrosion resistance. The competing aluminum alloys with high mechanical resistance from the 2000 and 7000 series have low corrosion resistance. Results of research on phase evolution, microstructure analysis, and the corrosion resistance of the as-cast and heat-treated alloy were presented. Microstructural analysis of the alloy was performed before and after the corrosion test.

## 2. Materials and Methods

Optimisation of modelling criteria values was performed by the Pareto multi-objective module from MATLAB software (MathWorks, Natick, MA, USA, v 6.02) and by the IMNR software developed for the calculation and sorting of the optimisation results.

The thermodynamics, multicomponent phase equilibrium, multi-phase precipitation kinetics, and nonequilibrium solidification were studied using MatCalc Pro edition (MatCalc Engineering GmbH, Vienna, Austria, v 6.03) with aluminum alloys databases ME-Al1.2 (thermodynamic) and ME-Al_rel1.0e (kinetic).

The proper alloy compositions were obtained using an induction furnace type Linn MFG-30 (Linn High Therm GmbH, Eschenfelden, Germany), equipped with an inert atmosphere and cast in a copper mould. The alloy charge was 250 g of technical purity raw materials of Al, Cu, Si, Zn, and Mg, placed in an alumina-based crucible. The melting process reached over 700 °C to allow the dissolution of all the elements. The resulted ingot proceeded to heat treatment in an LHT 04/17 Nabertherm GMBH (Lilienthal, Germany) electrical furnace under a protective atmosphere (Ar). The annealing stage was conducted at 400 °C for 30 h, with slow heating and cooling stages of 0.1 °C/s and 0.03 °C/s, respectively.

An inductively coupled plasma spectrometry (ICP-OES) of type Agilent 725 spectrometer (Santa Clara, CA, USA) was used to study the chemical composition of the alloys. To determine the most accurate values, there were taken samples from different areas of the alloy ingot to be investigated. Optical micrographs were obtained with a Zeiss Axio Scope A1m Imager microscope (Jena, Germany), with bright field, dark field, DIC, and polarization features. To increase the grains’ visibility and boundaries, the ingot samples were previously etched using a Keller solution. An FEI Quanta 3D FEG microscope (FEI Europe B.V., Eindhoven, The Netherlands) that operates at 20–30 kV and is equipped with an energy dispersive X-ray spectrometer (EDS) was used to analyse samples by scanning electron microscopy (SEM). The configuration of the phases was studied with an X-ray diffractometry (XRD) (FEI Europe B.V., Eindhoven, The Netherlands). Data were acquired using a BRUKER D8 ADVANCE diffractometer (Bruker Corporation, Billerica, MA, USA), equipped with DIFFRACplus XRD Commender (Bruker AXS) software (v 2018, Bruker Corporation, Billerica, MA, USA), Bragg-Brentano diffraction method, Q–Q coupled in a vertical configuration. The following parameters were used: CuK_radiation, 2Q Region: 20–1240, 2Q Step: 0.020, Time/step: 8.7 s/step, speed of rotation is 15 rot/min. Cuk_ radiation was removed by a SOL X detector (Bruker Corporation, Billerica, MA, USA). Obtained data were processed with Bruker^®^ Diffracplus EVA Release 2018 software to identify the ICDD^®^ Powder Diffraction File (PDF4+, 2019 edition) database and the full pattern appropriate (FPM) module of the same software package.

To determine the Compression strength for the obtained alloys, we used an LBG testing machine, model TC-100 (LBG Testing Equipment SRL, Azzanos, Paolo, Italy), with a maximum load of 100 kN. The analyzed samples were alloy bars with a diameter of 6 mm and a length of 6 mm. The speed of compression was established at 6 mm/min.

To determine the Vickers microhardness, the alloys were analyzed at ambient temperature using microindenter attachment (Anton Paar MHT10, Anton Paar GmbH, Graz, Austria), which has an applied load of 2 N and slope of 0.6 N/s. To identify the average values for each sample, 10 measurements were made. The samples obtained before and after the annealing process were evaluated in terms of corrosion resistance by the linear polarization resistance technique.

The anticorrosive properties of the coated electrodes were determined in 3.5 wt% NaCl solution solutions, using the potentiodynamic polarization and electrochemical impedance spectroscopy (EIS, Lyon, France) methods. The corrosion resistance was determined on Voltalab 80 PGZ 402 equipment (Radiometer Analytical SAS, Lyon, France), equipped with a Volta Master 7.0.8 software. To prevent electrical interferences, the cell was installed in a Faraday cage.

A thermostatic glass cell was used as an electrochemical cell. It was composed of three electrodes: the counter electrode was made of Pt (wire 2 mm), the reference electrode was calomel, which was saturated with KCl (SCE/KCl (sat)), and the working electrodes were: P-115 cast, P115-heat treated and for comparison A570 Gr 40 steel (S = 0.5 cm^2^). The standard composition of the A570Gr40 steel was: 99% Fe, 0.25% C and 0.20% Cu, 0.9% Mn, 0.04% P and 0.05% S as impurities. Minimal sample preparation consisted of placing the electrodes on double-sided carbon tape with no coating. Before measurement, the working electrodes were burnished with several emery paper sheets of various grades (250 up to 4000) until a mirror-like surface was reached. After the polishing process, the electrodes were swilled with acetone and bidi stilled water and then were dried at ambient temperature and introduced into the electrochemical cell.

Corrosion was evaluated by the polarization method (OCP, linear polarization, and Tafel plots). Expanding the potential from the cathodic to the anodic potential in the interval between −1500 mV to +250 mV (SCE) at a scanning rate of 2 mV/s is useful to explore the Tafel polarization curves.

Measurements of Electrochemical impedance spectroscopy were realized in frequency intervals between 100 kHz to 0.01 Hz with an amplitude of 10 mV to an open circuit potential of the working electrodes in the study.

## 3. Results

### 3.1. Selection of the Alloy Composition

Several simulation and criteria calculations were performed for the Al-Cu-Si-Zn-Mg system. The criteria and calculation methods are presented previously in [13]. The data obtained for the same alloy system in [13] was further developed with a larger number of calculations and approximations with the help of the MATLAB optimisation module and internally developed software for the calculation and sorting of the most appropriate compositions. Regarding the choice of the adequate composition, calculations based on the Hume-Rothery criterion were performed, according to which almost any unidentified composition can be predicted because the necessary thermophysical parameters can be easily reached [14,15].

It was very important to identify the most appropriate composition of the complex, concentrated alloy, characterised by good endurance in a corrosive marine environment. The influence of the alloy composition is very strong because the alloying elements have a high impact on the properties of CCA. The presence of aluminium in the alloy has a major influence on the evolution of the phase structure. The aluminium properties, like ductility, good mechanical resistance, and corrosion resistance, are also important in element selection. In addition, this element is most often found in the lightweight complex concentrated alloys, along with other metals like Si, Mg, Ti, and B. As fragility is not a desired property in the types of applications pursued in this study, an attempt was made to avoid it by choosing elements such as Cu and Zn, which can improve the material machinability.

### 3.2. CALPHAD and Kinetic Simulation

The thermodynamic simulation results for Al_5_Cu_0.5_Si_0.2_Zn_1.5_Mg_0.2_ alloy are presented in Figure 1 and Figure 2 and show a predominant presence of solid solutions (FCC-A1 and BCC-A2), in the alloy structure. The intermetallic Al_2_Cu phase is also present in notable proportion at low temperatures. Several other intermetallic phases (Zn_2_Mg and Mg_2_Si) are also identified, but in minor proportions.

The nonequilibrium solidification behaviour (Figure 2) shows the same solidification order for the component phases. The phases with termination S represent the cumulative solidification patterns, containing equilibrium and nonequilibrium values. The first phase to form during solidification is FCC-A1, followed by BCC-A2 and Si-A4. The Al_2_Cu phase is not shown in the Scheil diagram as it is forming at lower temperatures.

Figure 3 shows the results for the simulation of precipitation of intermetallic phases during the annealing process. To present the as-cast structure at the beginning of the annealing stage, the system was set up to start with the solidification of the alloy after the preparation process (Figure 3a). The solidification process was simulated according to the furnace characteristics as the alloy was cast in a copper mould, at about 800 °C, with rapid solidification up to 400 °C. A lower rate was then involved in cooling to room temperature. The heating and cooling rates for the annealing process were very low to avoid phase interface cracks due to dissimilar thermal expansion coefficients. The precipitation behaviour of the intermetallic phases indicates, as expected, a high fraction of the Al_2_Cu phase followed distantly by the Mg_2_Si phase (Figure 3b). Very low proportions of the Zn_2_Mg were identified. The precipitation number count and size (Figure 3c,d) show different behaviour from the as-cast and annealed state, as the minor intermetallic phases (Mg_2_Si and Zn_2_Mg) have much closer values with the Al_2_Cu phase. This is due to the less stable character of the minor intermetallic phases, present in higher proportion in nonequilibrium as-cast structures.

### 3.3. Experimental Results

#### 3.3.1. Chemical Analyses

The nominal and experimental composition of the alloy are presented in Table 1. It can be observed that the values are very close, the differences between the two compositions being around 2 wt%. Unlike the other elements of the alloy composition, aluminum is an exception, for which it is registered as a nominal composition, which exceeded by 5 wt% the experimental composition. Considering that the alloy elements are found in a high percentage, the variation of 2 wt% in two types of compositions does not significantly affect the structural integrity of the alloy.

#### 3.3.2. Microstructure Analyses

The optical micrographs of the analysed alloy (Figure 4) showed substantial differences regarding the as-cast and heat-treated sample morphologies. The optical images for the as-cast alloy revealed a fine and well dispersed dendritic microstructure (Figure 4a). Several constituent phases were also observed in the interdendritic area, including a Chinese script morphology eutectic and a punctiform and lamellar eutectic. The heat-treated sample (Figure 4b) also shows a dendritic structure with a large interdendritic area. The four interdendritic phases are present in the form of platelets trapped in the main phase matrix. The eutectic structure was also identified in the interdendritic area.

Scanning electron microscopy analyses (Figure 5) show a substantial difference between the as-cast and annealed states. The as-cast alloy (Figure 5a) was characterized by multiple phases homogeneously arranged in the material. Thus, two distinct main phases can be identified; one of these is constituted by dendrites, and the second phase is placed in the interdendritic area. As a result of heat treatment, it can be distinguished by a large dendritic alloy structure and clearly defined eutectic structures (Figure 5b). Four well-distributed phases were identified by scanning electron analysis. One of the phases has an acicular shape, while three of them occurred in the form of platelets. The composition of Al_5_Cu_0.5_Si_0.2_Zn_1.5_Mg_0.2_ phase structure in as-cast and annealed states are presented in Table 2 and Figure 6. According to EDS analyses of the as-cast alloy, a high concentration of Al and Zn in the dendritic area (DR) was spotted. The interdendritic area (ID) consists mainly of Cu, which along with Al, Zn, and Mg, forms well-defined phases (ID1, ID2, ID3, and ID4). The eutectic structures, investigated with EDS mapping (Figure 6 and Figure 7), show a basic composition consisting of Al and Zn. It can be observed that Mg and Si are concentrated in common material regions, constituting intermetallic compounds.

The diffraction patterns of the Al_5_Cu_0.5_Si_0.2_Zn_1.5_Mg_0.2_ alloy in as-cast condition (Figure 8) indicate a structure composed mainly of two solid solution phases (A1-Al with reference PDF No: 01-077-6849 and A3-Zn with reference PDF No: 01-078-9363) and Al_2_Cu intermetallic (PDF No: 04-001-0923). Complex and less stable Mg_8_Cu_2_Al_4_Si_7_ (PDF No: 04-009-1416) and Mg_2_Zn_11_ (PDF No: 04-007-1412), and Si (PDF No: 00-026-1481) were also detected in the as-cast structure of the alloy. The phase count has changed in the annealed state (Figure 9), showing a less stable formed phase Al_4.2_Cu_3.2_Zn_0.7_ (PDF No: 00-047-1393). The X-ray analysis showed that the crystal structure is almost identical for the as-cast and heat-treated alloy, with sharp, high peak intensities for FCC-A1 and HCP-A3 between 38–45°. Knowing that Al and Zn have the highest atomic proportions in the alloy, the high peak intensities for FCC-A1 and HCP-A3 also show a possible high proportion of these two phases in the alloy mass. There is no significant difference between the as-cast and heat-treated alloy samples regarding peak intensities, showing the high stability of the alloy structure.

#### 3.3.3. Mechanical Tests

Characteristic diagrams of the compression tests are presented in Figure 10. The graph analysis shows that the yield strength of the material is between 500 and 600 MPa, and the ultimate strength is between 800 and 900 MPa. These values are considered superior to most aluminium alloys. Plastic deformation behavior was observed for the studied alloy. The deformation slope has a linear ascending aspect in the plastic range, finishing with a descending rupture curve at approximately 800 MPa compression strength and 0.04 compression strain. The material does not show a sudden brittle rupture which allows for use in various complex applications, where high strength and low density are required.

HEA alloy samples (Al_5_Cu_0.5_Si_0.2_Zn_1.5_Mg_0.2_) with low density were tested for micro hardness and mechanical strength.

The results of Vickers microhardness tests are presented in Table 3. The results showed that the as-cast samples have higher microhardness than the annealed specimen. This is mostly due to the phase morphology, which has changed from sharp dendrites to a more rounded phase configuration. The microhardness range is similar to the 2000 and 7000 series aluminium alloys.

#### 3.3.4. Corrosion Tests

The corrosion resistance of the samples was analyzed by potentiodynamic polarization measurements (linear polarization resistance (LPR), Tafel plots) and electrochemical impedance spectroscopy. Tests were performed in aerated 3.5 wt% NaCl solution. The polarization of A570 Gr 40 carbon steel was studied for comparison.

Polarization curves are presented in Figure 11, while Table 4 presents the suitable kinetic parameters (corrosion potential-E_corr_, and corrosion current density-i_corr_), as well as the calculated corrosion parameters: polarization resistance -R_p_ and corrosion rate -CR (determined by Tafel’s extrapolation method).

The E_corr_ values for the as-cast and heat-treated samples were very close to each other, −1.014 V_SCE_ and −0.959 V_SCE,_ respectively, significantly lower than the E_corr_ value for the steel sample (−0.687 V_SCE_). The i_corr_ values for as-cast (7.4 × 10^−5^ A) and heat-treated (2.1 × 10^−5^ A) samples are an order of magnitude smaller than the steel sample (1.4 × 10^−4^ A). These large differences result in a significant improvement of the overall corrosion rate for the selected alloy in either obtaining state. There is also a significant difference between the corrosion results of the two alloy samples, as the heat-treated state indicates twice as good corrosion resistance. The corrosion rate for both studied alloys proved to be significantly lower than steel (0.6984 mm/year). As expected, the corrosion rate of the heat-treated binder (0.1972 mm/year) is lower than that of the as-cast alloy (0.3424 mm/year), which demonstrates that the heat treatment improves the corrosion resistance of the alloy. The passivation regions for all the samples are forming late, at approx. 0.1 A/cm^2^. This suggests the formation of a thin or penetrable oxide layer at the alloy surface. Overall, the results show a good corrosion resistance for the Al_5_Cu_0.5_Si_0.2_Zn_1.5_Mg_0.2_ alloy, compared to A570 Gr 40 steel, and are in good agreement with the polarization resistance obtained by LP. The alloy impedes the attack of the aggressive ions (Cl^−^) on the electrode surface. As the above data shows, the best corrosion rate was obtained for the heat-treated alloy.

Electrochemical impedance spectroscopy (EIS) provides additional specific information on the corrosion behavior of CCA steel and alloy samples. The EIS experimental tests were performed for the open circuit potential (OCP), on the frequency range from 100 kHz to 40 MHz, with an AC wave of ±10 mV (peak-to-peak). The impedance spectra of the samples, analysed in a solution of NaCl, are shown in Figure 12. The electrochemical impedance spectra are presented by a semicircle, with a high-frequency capacitance loop and a low-frequency inductive loop. However, these capacitive loops are not perfect semicircles, which is attributed to the frequency scattering effect due to the roughness and inhomogeneity of the metal surface.

Mainly, the capacitive loop in the Nyquist diagrams assumes a time constant, correlated with the charge transfer process of the corrosion compounds on the electrode surface. From the analysis of the impedance spectra of the samples, it can be seen that the diameter of the semicircles varies with the evolution of the corrosion process due to the presence of the film composed of corrosion compounds.

The impedance modulus, at low frequencies, increases with the increase of the corrosion formed by the corrosion compounds, and an increase in Z_mod_ denotes a higher protection capacity. Consequently, the heat-treated Al_5_Cu_0.5_Si_0.2_Zn_1.5_Mg_0.2_ alloy has better corrosion resistance in 3% NaCl solution than the as-cast specimen regular steel, and the heat-treated specimen provides even better resistance than the as-cast specimen and steel.

The Bode diagrams (Figure 13) show a single time constant, suitable for a well-established phase angle of approx. 38° for steel, 47° for cast alloy, and 58° for the heat-treated alloy, which reveals that at high frequencies have a capacitive behaviour, and at low frequencies, they have an inductive behaviour, with a low diffusive tendency. An equivalent circuit was presented for the obtained EIS data. To provide an accurate fit, the phase element constant (CPE) was introduced instead of the usual pure double-layer capacitor (Cdl). The significance of CPE is the deformation of the capacitive semicircle, showing the heterogeneity of the corroded surface. The CPE impedance can be defined as: Z_CPE_ = Y_0_ − 1 (jω)^−n^, where ω is the angular frequency, j is the imaginary number (j^2^ = −1), Y_0_ is the amplitude comparable to capacitance, and *n* is the phase change. The phase shift value provides details on the degree of inhomogeneity of the metal surface. The higher the value of *n*, the lower the surface roughness, i.e., a reduced surface inhomogeneity. CPE can be established as resistance when *n* = 0, (Y_0_ = R), capacitance when *n* = 1 (Y_0_ = C) and inductance when *n* = −1 (Y_0_ = 1/L) or Warburg impedance when *n* = 0.5 (Y_0_ = W) based on the value of *n*.

The analysis of the experimental data was performed by processing the results at the appropriate equivalent circuit shown in Figure 14 and the various impedance characteristics such as solution resistance (R_s_) and load transfer resistance (R_ct_). L_1_ and R_3_ are inductive elements. The double-layer capacity (Cdl) was calculated along with other parameters and indicated in Table 5.

The inductive behaviour present in the low-frequency domain is due to the relaxation process of some added species (corrosion compounds) to the surface of the working electrode. It is a process of adsorption on the surface of the electrode. The results obtained from the EIS tests indicate that the load transfer resistance Rct has increased, and the Cdl double layer capacity has been reduced with the consolidation of the film from protective corrosion compounds.

The EIS measurements results confirm the polarization data obtained from OCP, LP, and Tafel, respectively, that the heat-treated alloy has the best corrosion resistance compared to cast alloy and steel.

The microstructural characterisation results of the corroded films are presented in Figure 15 and Figure 16. It was generally distinguished as an irregular structure with various phases of different morphologies. The corrosion layers investigated on both samples appeared to be partially fractured either due to the chemical attachment of the testing solution or due to the sample preparation process. There are no large cracks between the corrosion film areas. The layers’ appearance showed a significantly finer morphology for the heat-treated sample. The EDS analysis results, presented in Table 6, show many phases with different compositions. As expected, oxygen is predominant in most of the phases. Al and Zn were also found as the main elements. Nevertheless, there are some areas in the corroded films that have higher concentrations of Cu and Si. In general, Si presence in a higher percentage determines a lower oxygen content, as identified in phases 2 and 6 from the as-cast sample and phases 3, 6, and 8 from the heat-treated sample.

## 4. Discussions

The experimental results show a relatively complex structure with one main dendritic phase and several interdendritic phases. The SEM-EDS and XRD investigations identified 5 interdendritic phases for the as-cast alloy and 6 phases for the annealed alloy. The main dendritic phase is FCC-A1, with a large representation in the alloy structure. The heat-treated sample contains, in addition, an Al_4.2_Cu_3.2_Zn_0.7_ phase (t′ phase), which shows to be stable in the annealed structure. The EDS and XRD analyses seem to illustrate the same type of phases as the element proportions are similar to the lattice type.

From the comparison of the modelling and experimental results, it can be observed that the number of phases agrees between the as-cast sample and CALPHAD findings, but have a reduced agreement on the type of phases. The BCC-A2, Zn_2_Mg, and Mg_2_Si phases, obtained by the simulation process, were replaced by HCP-A3, Mg_8_Cu_2_Al_4_Si_7,_ and Mg_2_Zn_11_ in the experimental samples. The Al_4.2_Cu_3.2_Zn_0.7_ phase was not predicted by the simulation process.

The solidification behaviour simulated with the Sheil-Gulliver method through the MatCalc software shows that the first phases to form are FCC-A1 (aluminum-based), BCC-A2, and Si-A4. The dendrite formation identified in the optical and SEM-EDS analyses seems to have a composition with a large Al proportion, suggesting the FCC-A1 type of phase, which was also identified in the XRD analyses. Dendrites are known to be the first phases to form in an alloy, so the modelling results and experimental results are similar in this regard. However, it is hard to tell what is the BCC-A2 phase in the experimental sample as the XRD results show an A3-Zn-based phase instead.

The simulation of precipitation kinetics showed a stable Al_2_Cu intermetallic phase, doubled by Mg_2_Si, but at a significantly lower concentration. The Mg_2_Si was not identified either in the as-cast or heat-treated samples, which demonstrates the high difficulty in predicting phase structures in complex concentrated alloys. The Al_2_Cu intermetallic phase was also identified by the XRD analyses and presented the suggested composition for the ID1 phase in SEM-EDS analyses. As in the simulation results, the ID1-Al_2_Cu phase shows smaller dimensions in the heat-treated state than in the as-cast state.

The mechanical resistance tests showed relatively high values compared to the conventional aluminum alloys from the 2000 and 7000 series. The complex structure of the alloy and the presence of the Al_2_Cu intermetallic phase contributes significantly to the high values obtained at mechanical resistance tests. Moreover, the corrosion resistance shows a substantial improvement versus regular steel. Al_5_Cu_0.5_Si_0.2_Zn_1.5_Mg_0.2_ presented a corrosion rate six times slower than the steel sample. The impedance corrosion results also pinpoint a significant improvement in corrosion resistance for the studied complex concentrated alloy. Due to the structural, mechanical, and corrosion results obtained, the developed alloy provides the required characteristics for applications in an offshore environment where lightweight, high mechanical resistance, and good marine water corrosion resistance are required.

## 5. Conclusions

A new lightweight complex concentrated alloy, composed of more common elements, is studied in the paper. The behaviour of the alloy in the as-cast and heat-treated states was presented in terms of structural, mechanical, and corrosion resistance analyses. Simulation results with dedicated software were also presented and compared to the experimental findings. Simulation of the alloy structure performed by CALPHAD showed a structure composed mainly of solid solutions of FCC and BCC types. The main intermetallic phase suggested by the thermodynamic and diffusion simulation was the Al_2_Cu phase, in significant proportion at low temperatures. Complex structures were identified by: optical, SEM-EDS, and XRD analyses with a predominant dendritic phase containing Al in large proportion. The interdendritic phase that is shown to present a larger proportion is Al_2_Cu. The BCC type phase, determined by the simulation software, was replaced with a Zn-based A3 solid solution identified in the XRD analyses. Other complex intermetallic phases were shown to be stable in the studied alloy at room temperature (Mg_8_Cu_2_Al_4_Si_7_ and Al_4.2_Cu_3.2_Zn_0.7_). Si was found mostly segregated in the alloy structure. The mechanical characterization of the Al_5_Cu_0.5_Si_0.2_Zn_1.5_Mg_0.2_ alloy revealed high values for compression yield strength (500–600 MPa) and microhardness (238 HV) in the as-cast state. The annealing process had a softening effect, as expected. The determined values are comparable with the 2000 and 7000 series aluminum alloys.

The corrosion tests showed a relatively low value for the corrosion rate, either in as-cast (0.3424 mm/year) or heat-treated state (0.1972 mm/year). The passivation regions for all the samples are forming late, at approx. 0.1 A/cm^2^, which suggests the formation of a thin or penetrable oxide layer at the alloy surface. It is obvious that the alloy impedes the attack of the aggressive ions (Cl^−^) on the electrode surface. The best corrosion rate was obtained for the heat-treated alloy. Good corrosion resistance was also identified by the impedance tests in the annealed (6.5 ohm·cm^2^) and the as-cast samples (5.5 ohm·cm^2^). The inductive behaviour present in the low-frequency domain is due to the relaxation process of some added species (corrosion compounds) to the surface of the working electrode. It is a process of adsorption on the surface of the electrode. The results obtained from the EIS tests indicate that the load transfer resistance Rct has increased, and the Cdl double layer capacity reduced, with the consolidation of the film from protective corrosion compounds.

The characterisation results showed good potential for the novel alloy in applications that involve low density, high mechanical resistance, and good corrosion resistance.

## Figures and Tables

**Figure 1 materials-15-03345-f001:**
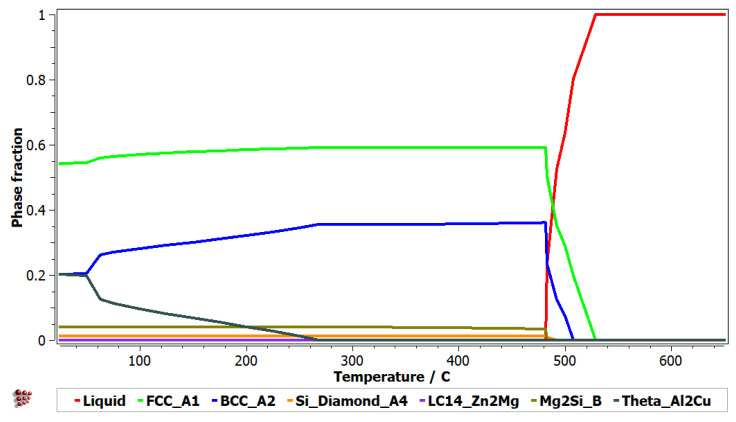
Simulated phase diagram for Al_5_Cu_0.5_Si_0.2_Zn_1.5_Mg_0.2_ alloy.

**Figure 2 materials-15-03345-f002:**
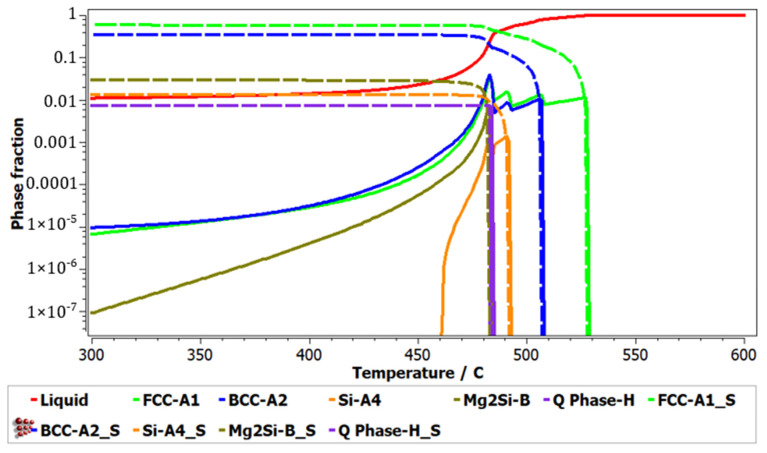
Simulated Scheil-Gulliver diagram for Al_5_Cu_0.5_Si_0.2_Zn_1.5_Mg_0.2_ alloy.

**Figure 3 materials-15-03345-f003:**
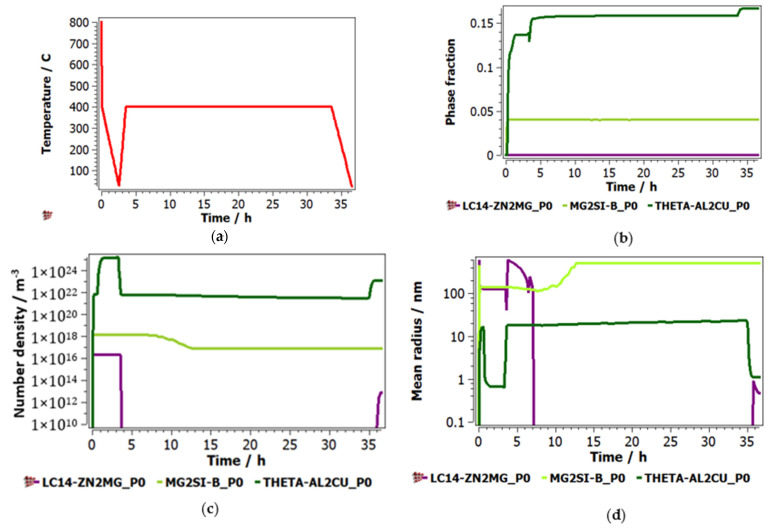
Simulated precipitation kinetics diagram for Al_5_Cu_0.5_Si_0.2_Zn_1.5_Mg_0.2_ alloy after the annealing process: (**a**) heat treatment diagram, (**b**) precipitate phase fraction versus time, (**c**) precipitate number density versus time, (**d**) precipitate mean radius versus time.

**Figure 4 materials-15-03345-f004:**
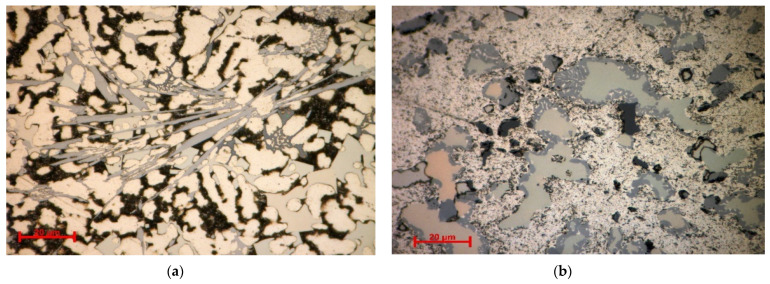
Optical micrographs of (**a**) as-cast and (**b**) annealed Al_5_Cu_0.5_Si_0.2_Zn_1.5_Mg_0.2_ alloy.

**Figure 5 materials-15-03345-f005:**
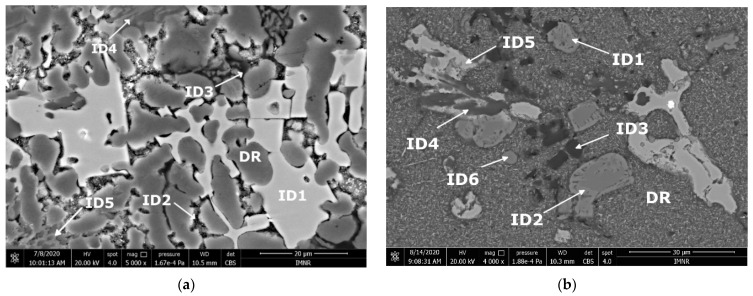
SEM images of (**a**) as-cast and (**b**) annealed Al_5_Cu_0.5_Si_0.2_Zn_1.5_Mg_0.2_ alloy.

**Figure 6 materials-15-03345-f006:**
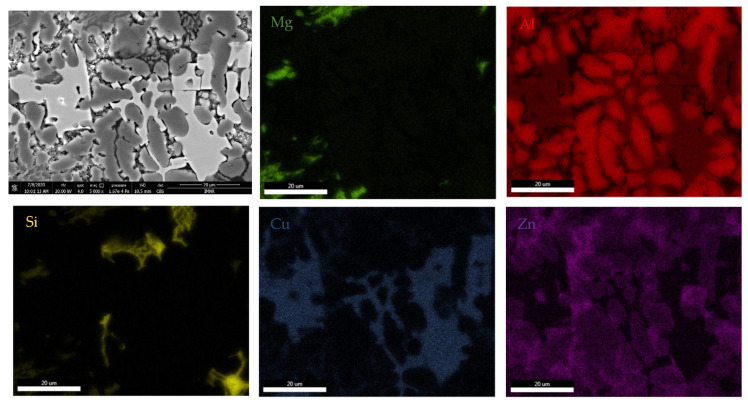
EDS mapping of the Al_5_Cu_0.5_Si_0.2_Zn_1.5_Mg_0.2_ as-cast alloy.

**Figure 7 materials-15-03345-f007:**
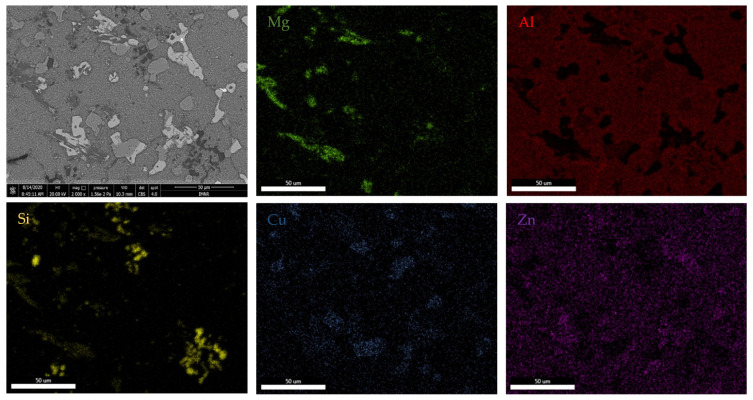
EDS mapping of the Al_5_Cu_0.5_Si_0.2_Zn_1.5_Mg_0.2_ heat treated alloy.

**Figure 8 materials-15-03345-f008:**
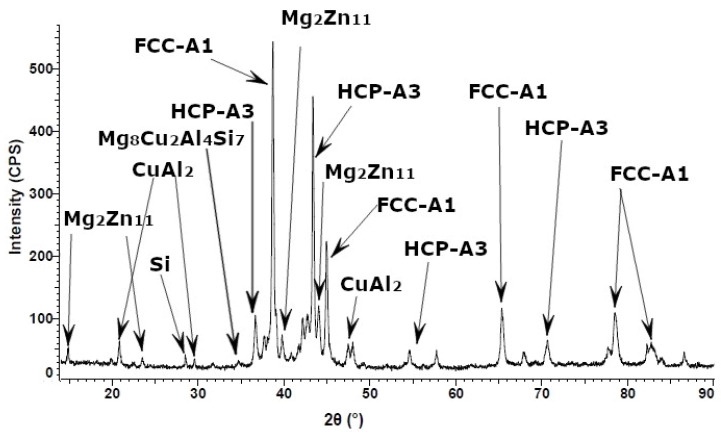
X-ray diffraction pattern for as-cast Al_5_Cu_0.5_Si_0.2_Zn_1.5_Mg_0.2_ alloy.

**Figure 9 materials-15-03345-f009:**
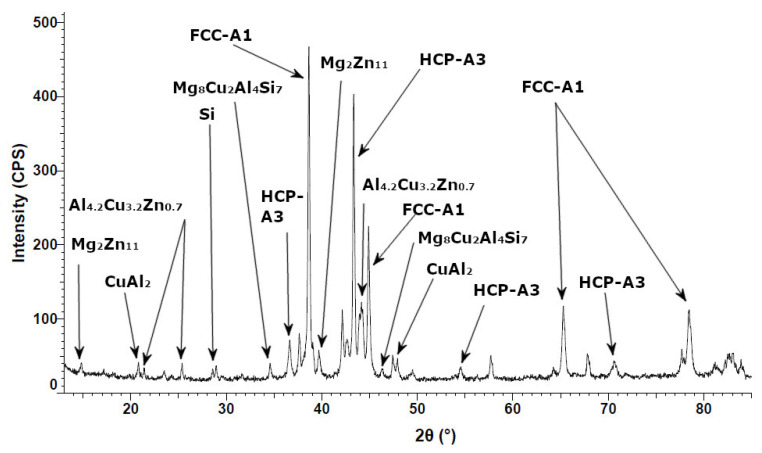
X-ray diffraction pattern for heat-treated Al_5_Cu_0.5_Si_0.2_Zn_1.5_Mg_0.2_ alloy.

**Figure 10 materials-15-03345-f010:**
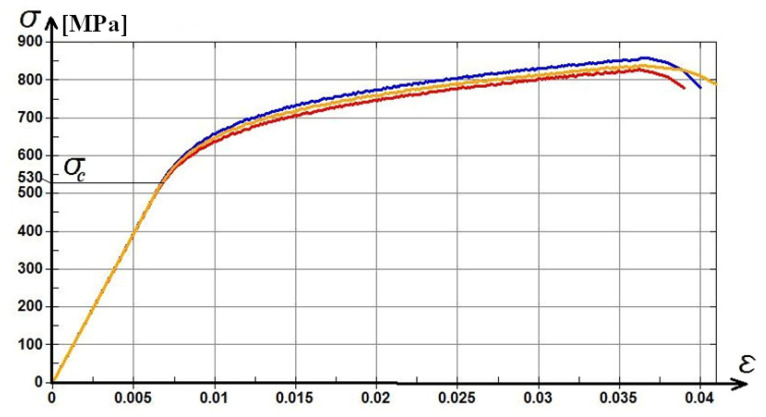
Strength-strain diagram of compression tests performed on as-cast sample (σ—compression strength, σ_c_—yield strength, and ε—strain).

**Figure 11 materials-15-03345-f011:**
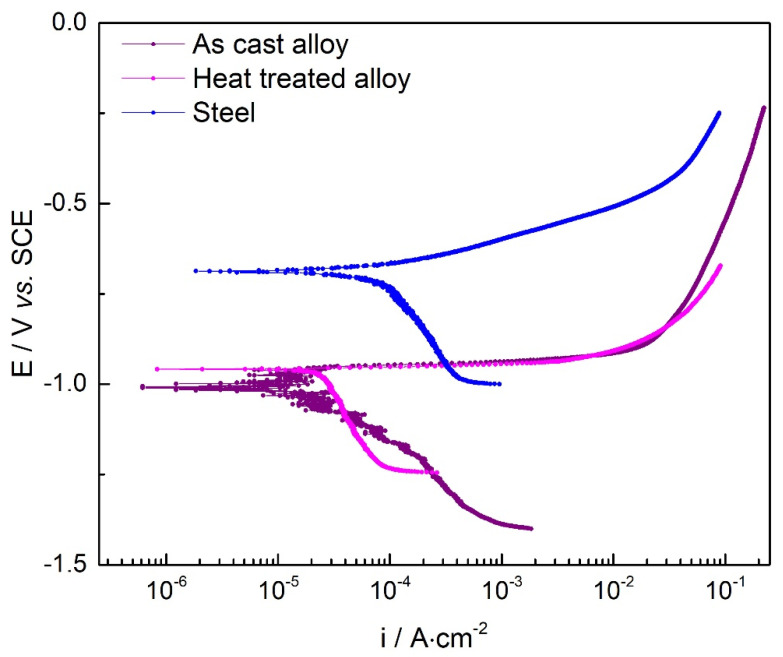
Tafel polarization curves for the as-cast alloy, heat treated alloy and steel.

**Figure 12 materials-15-03345-f012:**
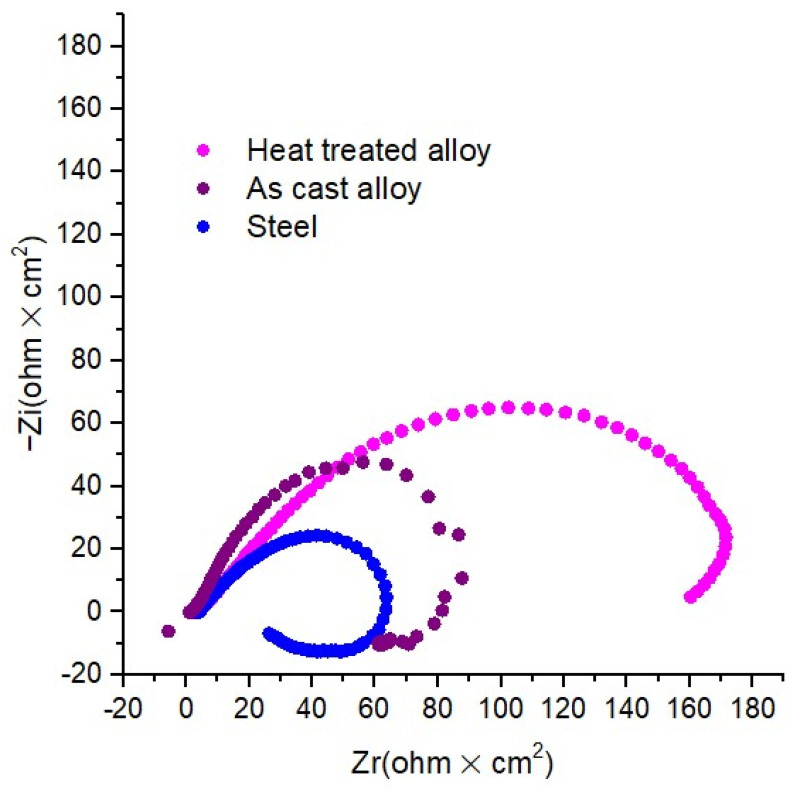
Nyquist plots for the as-cast alloy, heat-treated alloy, and steel.

**Figure 13 materials-15-03345-f013:**
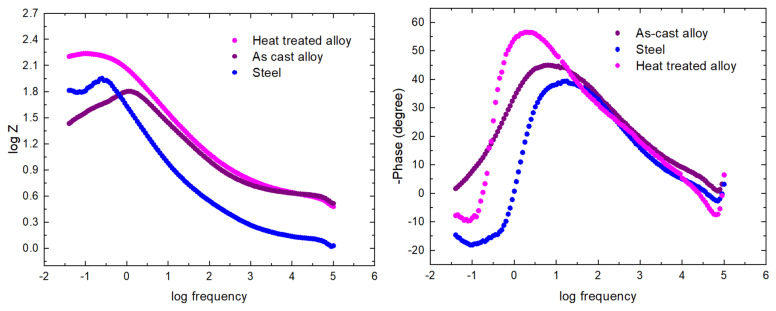
Bode plots for the as-cast alloy, heat-treated alloy, and steel.

**Figure 14 materials-15-03345-f014:**
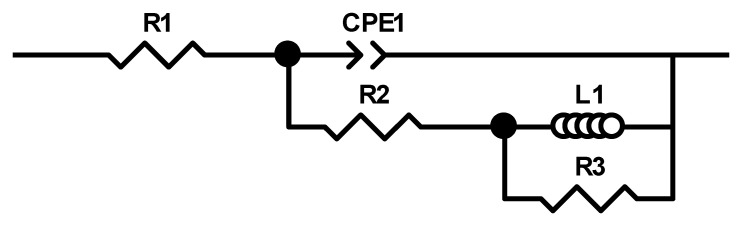
Equivalent circuit for the impedance tests.

**Figure 15 materials-15-03345-f015:**
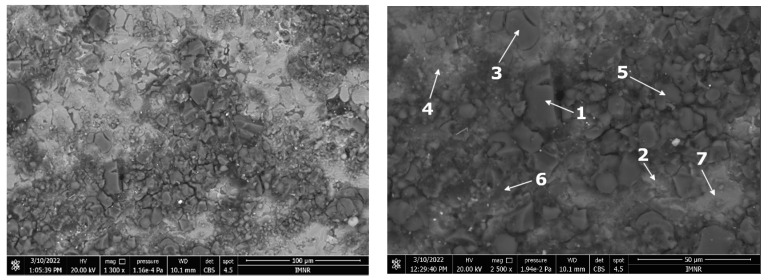
BSE-SEM images of the corroded surface of the as-cast alloy sample after the impedance tests. The marked areas were studied for EDS composition in Table 6.

**Figure 16 materials-15-03345-f016:**
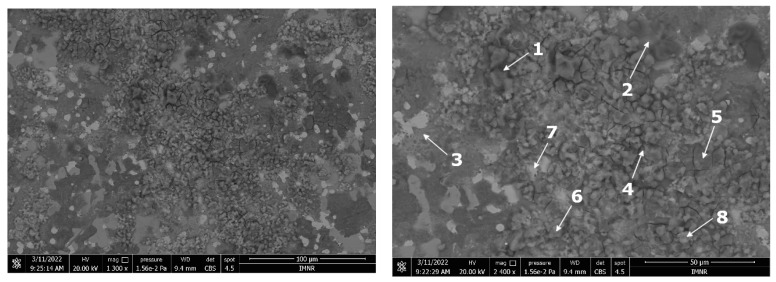
BSE-SEM images of the corroded surface of the heat-treated alloy sample after the impedance tests. The marked areas were studied for EDS composition in Table 6.

**Table 1 materials-15-03345-t001:** Chemical composition of the alloy expressed in weight percentage.

Type of Composition	Al	Cu	Si	Zn	Mg
Nominal	49	11.55	2	35.6	1.8
Experimental	45.48	11.49	2.08	38.87	2.05

**Table 2 materials-15-03345-t002:** EDS composition for the Al_5_Cu_0.5_Si_0.2_Zn_1.5_Mg_0.2_ alloy, in at.%.

State	Phase	Composition, at.%				
		Al	Cu	Si	Zn	Mg
As-cast	DR	82.79	2.51	1.83	11.38	1.49
	ID 1	63.41	30.75	2.26	1.95	1.62
	ID 2	68.02	7.60	2.60	18.99	2.77
	ID 3	22.53	1.26	71.77	3.84	0.58
	ID 4	39.12	7.38	22.49	4.48	26.51
	ID 5	15.76	8.57	5.68	49.35	20.63
Heat-treated	DR	82.66	1.68	-	15.65	-
	ID 1	66.96	33.04	-	-	-
	ID 2	55.86	8.54	-	34.6	-
	ID 3	7.87	-	90.57	1.56	-
	ID 4	40.45	10.27	30.35	-	38.93
	ID 5	18.37	13.75	-	46.26	21.62
	ID 6	48.65	34.27	-	12.34	4.64

**Table 3 materials-15-03345-t003:** Microhardness results for the obtained alloy.

Specimen	HV
as-cast	238
annealed	168

**Table 4 materials-15-03345-t004:** The corrosion parameters of the tested samples in 3.5 wt% NaCl at 25 °C.

Samples	E_OCP_ (V)	R_p_ (Ω)	E_corr_ (V)	i_corr_·10^−4^ (A/cm^2^)	CR (mm/Year)
OL44	−0.589	11.76	−0.687	1.416	0.6984
as- cast	−0.986	554.20	−1.014	0.741	0.3424
heat treated	−0.995	2119.44	−0.959	0.213	0.1972

R_p_ = polarization resistance; E_corr_ = corrosion potential; i_corr_ = corrosion current density; CR = corrosion rate.

**Table 5 materials-15-03345-t005:** Impedance test results parameters.

Sample		CPE1							
	R_s_ ohm·cm^2^	Q-Yo S·s^−n^·cm^−2^	Q-n	R_ct_ ohm·cm^2^	L (H·cm^2^)	R_L_ ohm·cm^2^	χ^2^	R_p_	Cdl (μF·cm^−^^2^)
steel	0.6694	0.01775	0.6083	31.06	34.48	3.622 × 10^10^	6.791 × 10^−3^	67.949	521
As-Cast alloy	1.621	0.006374	0.6195	44.99	5.885	30.51	4.153 × 10^−3^	77.121	620
Heat treated alloy	2.103	0.004072	0.6201	99.38	14.84	28.96	3.971 × 10^−3^	130.443	815

Notation signification: R_s_—solution resistance, R_ct_—the charge transfer resistance, CPE—phase element constant, L, R_L_—inductive elements.

**Table 6 materials-15-03345-t006:** Phase composition for the corroded layer of the as-cast and heat-treated samples.

State	Phase	Composition, at.%							
		Al	Cu	Si	Zn	Mg	O	Na	Cl
As-cast	1	26.46	-	-	5.37	-	66.09	1.05	1.03
	2	39.55	4.06	8.43	6.32	10.02	31.41	-	0.20
	3	29.16	0.87	-	9.57	0.73	56.89	2.32	0.46
	4	48.82	2.64	-	10.67	1.13	35.99	0.75	-
	5	24.23	-	0.32	3.30	0.34	70.24	1.00	0.57
	6	31.96	3.14	3.28	3.98	6.11	48.62	1.76	1.15
	7	79.61	1.92	0.87	11.13	-	6.47	-	-
Heat treated	1	20.73	0.49	3.33	5.98	1.38	67.61	0.21	0.26
	2	33.19	15.97	-	5.28	-	38.95	6.34	0.27
	3	72.42	1.78	5.76	19.71	-	-	0.33	-
	4	15.40	19.60	-	27.07	1.41	29.20	4.25	3.07
	5	31.66	1.67	-	11.91	1.34	49.71	2.12	1.59
	6	39.17	6.01	8.66	25.51	10.08	8.61	1.01	0.95
	7	24.68	0.77	3.03	11.48	1.79	55.77	0.20	2.27
	8	21.58	6.72	7.78	23.98	5.76	30.24	2.86	1.08

## Data Availability

Not applicable.

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
