# Peer review of "Characterisation of a Novel Complex Concentrated Alloy for Marine Applications"

_materials, 2022, doi:10.3390/ma15093345_

Round 1

Reviewer 1 Report

The work is interesting and the scope of the research is very ambitious. The authors focused primarily on microstructural studies and the analysis of corrosion properties. Nevertheless, the authors did not avoid mistakes.
1. On what basis do the authors write that the analyzed material is classified as a light alloy? The high concentration of aluminum (45.5%) does not make it a light alloy, especially in the presence of high concentrations of copper and zinc. Line 437.
2. The authors willingly compare the alloy with aluminum alloys (2000 and 7000), however, when analyzing the obtained results of mechanical properties, it is difficult to classify these materials into the same group. The presented results of the research on mechanical properties are very poor. The authors write about the results of the compression and stretching test. However, charts for the tensile test only for the compression test are not shown. In the case of structural materials such as aluminum alloys, not only strength is important, but also ductility. The authors in lines 275, 276 write with high impact strength. Why such conclusions, since the value of the relative deformation under compression was 0.04? The results of the research on mechanical properties are described very vaguely! Why are the tensile test results not shown? It is worth rewriting this chapter.
3. Since the authors believe that the alloy is characterized by good toughness, please present the results of fractographic analyzes.
4. There is no information in the "material and methods" chapter for information on the mass of the ingot and the temperature of the melt (the information is only given in the results).

Reviewer 2 Report

The authors designed a new complex concentrated alloy and studied the material properties by chemical, structural, and mechanical in detail. However, the corrosion investigation is not as professional as that being done with the material studies. The figures were not well designed and the fitting results can not make sense. The detail comments can be seen below. Based on the review above, I do not recommend the publication without an major revision.

  1. Line 223: What is basis for this view point?
  2. Line 136: What is the background of the application of this material? What equipment will this material be used for? Why 3.5% NaCl solution was tested as the solution?
  3. Line 154: The EIS used a frequency ranging from 100 kHz to 0.04 Hz in the manuscript. why is not 0.01 Hz as the low frequency?
  4. Line 293: Why is the carbon steel selected for comparison?
  5. Figure 11: The number 1,2,3 should be labeled in the image. However, there is no need to label them.
  6. Table 4: The parameters can not match the image. The corrosion rate of steel is obvious less than the other materials, but the its Rp is the smallest. It does make sense.
  7. Figure 14: The scale and the range of X and Y axises should be same. Moreover, the fitted line should be provided. It is suggested that the lines use the same color as that in Figure 11.
  8. Line 360: As cast seems to be of different fitting loops. It should be described in detail.
  9. Line 430: It is unreasonable to say that due to the lack of corresponding results.
  10. More details on the corrosion behavior conclusions should be referred in the section of Conclusions.

Reviewer 3 Report

1. Overall, figures 8,9 and 10 lack detailed explanation in text. Also, the diagram in figure 10 should include the name of the parameters, instead just designations in Greek alphabet or the definition of the parameters should be added to text.

2. The statement "The best corrosion rate was obtained for the heat treated alloy" should be clarified - what is the value, is that a good feature or it shows that the heat treated alloy has worse corrosion resistance, authors should add more details and state the significance of this sentence. 

Reviewer 4 Report

Review of paper no. materials-1693373 titled Characterization of a novel complex concentrated alloy for marine applications by I.-C. Badea et al.

This is an interesting and well-researched paper that studies the microstructure, phase constitution, and corrosion behavior of an Al5Cu0.5Si0.2Zn1.5Mg0.2 alloy. The paper is publishable subject to revision.

1.The abstract should include concrete results, i.e., corrosion rates, phases observed, etc.

2.The experimental results (Sections 3.3.1 and 3.3.2) should be moved upfront and placed before the modelling results (Section 3.2).

3.Which database of thermodynamic parameters have you used for Calphad modelling? Please, state it in the methods section.

4.Table 2 should include the volume fractions of the microstructure constituents.

5.The XRD patterns (Figs. 8 and 9) should include the powder diffraction file numbers (PDF nos.) of the phases.

6.The authors compared the corrosion results with an OL44 steel. Please, specify the chemical composition of the steel.

7.The Al5Cu0.5Si0.2Zn1.5Mg0.2 alloy contains approximately 75 at.% Al. As such, the corrosion results should be compared with previously studied Al-TM alloys (TM=transition metal). See https://doi.org/10.3390/ma14185418 for a recent review and collection of Ecorr and jcorr values.

8.It would help to include the XRD patterns of the corroded alloys in the paper.

Round 2

Reviewer 2 Report

Fig. 12: The scale and the range of X and Y axises should be same. Moreover, the fitted line should be provided. 

Author Response

The scales of the X and Y axes were made different to accommodate the whole portions of the curves. Still, we modified the scale now and added new picture to the manuscript. Unfortunately, the  fitted lines cannot be imported from the equipment software as they are done automatically.

Reviewer 4 Report

This is a revised version of the previously reviewed manuscript. My comments have been only partially answered by the authors. Therefore, the paper still requires a revision before being reconsidered for possible publication in Materials.

1.The studied alloy (Al5Cu0.5Si0.2Zn1.5Mg0.2) has a complicated microstructure and phase constitution. It is necessary to explain it. The authors place their modelling results first, however, there is a disagreement between the modelling and experiment. Just a quick look at the database used by the authors (ME-Al1.2, https://www.matcalc-engineering.com/blog/wp-content/uploads/2020/04/ME-Al1.2_release-notes_update_200420.pdf) shows that it is optimized for Al-based alloys mainly. The present alloy had a considerable amount of Zn (39 wt. %) and Cu (11.6 wt. %, Table 1). As such, the used thermodynamic database had certain limitations, and some phases might have been omitted. The experimental observance of the phases (Figs. 8 and 9) need to be sufficiently discussed.

2.The physico-chemical properties of the alloy (mechanical properties, corrosion parameters) must be compared with materials of comparable chemical composition. The presented alloy and steel are incomparable in terms of chemical composition.

Author Response

Our answers to the new comments are:

  1. Even if the alloy has a considerable amount of Zn and Cu, the presence of Al in large proportion can offer close results as for aluminum alloys. The reviewer also mentioned that the studied alloy showed be compared with aluminum alloys in terms of corrosion resistance. There is still nowadays a limited thermodynamic data on the complex alloys or high entropy alloys. Even some specialized databases have been produced by some software vendors, known difficulties are met in simulating complex alloys and there are not always expected to fit the experimental results. Regarding figures 8 and 9 we have added supplementary observations in the revised manuscript.
  2. The most common material used in marine applications is steel, due to its very good mechanical properties. However, corrosion continue to be a challenge in marine environments. Aluminum alloys fight this problem very well, and in addition they have the advantage of being low-density alloys. The complex concentrated alloy has a The Al5Cu0.5Si0.2Zn1.5Mg0.2 alloy has superior mechanical properties compare to the 2000 and 7000 series aluminum alloys and better corrosion resistance than steel. The complex concentrate alloy combines the superior properties of conventional aluminum alloys and steels, achieving a material with low density, high mechanical resistance and good corrosion resistance.

We also added text in the introduction part that explains at the beginning why steel was chosen for corrosion resistance and why aluminum alloys were chosen for mechanical resistance.